# On demand shape memory polymer via light regulated topological defects in a dynamic covalent network

Wusha Miao[1], Weike Zou[1], Binjie Jin[1], Chujun Ni[1], Ning Zheng[1], Qian Zhao [1,2] & Tao Xie [1,2 ✉]

The ability to undergo bond exchange in a dynamic covalent polymer network has brought many benefits not offered by classical thermoplastic and thermoset polymers. Despite the bond exchangeability, the overall network topologies for existing dynamic networks typically cannot be altered, limiting their potential expansion into unexplored territories. By harnessing topological defects inherent in any real polymer network, we show herein a general design that allows a dynamic network to undergo rearrangement to distinctive topologies. The use of a light triggered catalyst further allows spatio-temporal regulation of the network topology, leading to an unusual opportunity to program polymer properties. Applying this strategy to functional shape memory networks yields custom designable multi-shape and reversible shape memory characteristics. This molecular principle expands the design versatility for network polymers, with broad implications in many other areas including soft robotics, flexible electronics, and medical devices.

[1] State Key Laboratory of Chemical Engineering, College of Chemical and Biological Engineering, Zhejiang University, 310027 Hangzhou, China. [2] ZJU-Hangzhou Global Scientific and Technological Innovation Center, 311215 Hangzhou, China. ✉email: taoxie@zju.edu.cn

Stimuli-responsive soft materials with tailorable properties are key enablers for numerous emerging applications including soft actuators[1–6], flexible electronics[7,8], and medical devices[9]. These properties are tied to the topological connectivity of covalent bonds, consequently locked into the materials during the synthesis/fabrication step. Dynamic covalent bonds, with their unique combination of adaptability and high bond strength, can be a paradigm shift. Indeed, dynamic covalent polymer network[10–12] exhibits adaptive properties not present in typical covalent networks such as self-healing, reprocessing, and permanent shape reconfiguration. In this context, two emerging opportunities are particularly noteworthy. The first is related to recycling of otherwise intractable thermoset polymers[13–22]. Classical thermoset polymers are widely used in high performance structural composites, but their chemically crosslinked nature makes them non-reprocessable. Triggering the dynamic characteristics of the covalent bonds, however, renders the network polymers reprocessable through covalent bond exchange within the network. The second opportunity is permanent shape reconfiguration in a mold-free manner via solid-state plasticity[23–26]. It allows access to complex shapes that are unobtainable with molding, an attribute particularly relevant to shape morphing structures/devices. Of importance in the current context is that the topological rearrangement in the above cases does not result in different topologies, consequently, the material properties remain identical before and after the dynamic bond exchange.

A counterintuitive opportunity arises when a dynamic covalent network is designed such that bond rearrangement leads to different topologies such as crosslinking densities, physical entanglements, and dangling chain length/distribution. For non-covalent supramolecular systems, achieving topological switching is more favorable given their more diverse connectivities[27]. However, supramolecular systems do not typically offer the mechanical robustness of their covalent counterparts. For dynamic covalent networks, topological transformation is typically accomplished by directing the network reconfiguration with external guiding molecules[28]. As elegant as the approaches are, the requirement for external molecules to participate the process limits it to non-crosslinked polymer solutions/polymer melts. The concept of topological isomerizable network (TIN)[29], by contrast, allows topological shifting within a "fully enclosed" solid material, that is, without participation of external reagents. This isomerization mechanism typically requires highly delicate design to introduce network heterogeneity. For instance, a TIN consisting of a permanent network frame and dynamic grafted long chains can isomerize from a grafting network into a brush network[29]. From a design standpoint, such a hybrid network (permanent mainframe and dynamic grafted chain) is quite specific and unusual. Two questions naturally arise: Is the TIN principle intrinsically narrow based or is it applicable to common and widely accessible polymer networks? From a material standpoint, what might be the unusual opportunities not offered by traditional approaches? To answer the first question, we realize that, besides inter-chain crosslinking, the universal topological features for real polymer networks are in fact a range of defects including intra-chain cycles (loops), dangling chains, and free chains (sol). These defects affect the macroscopic mechanical and thermomechanical properties in a non-constructive way, but are statistically unavoidable[30,31]. Consequently, minimizing these defects is often the goal in designing polymer networks.

In the present study, we aim to answer an intriguing question, that is, can the topological defects naturally present in real polymer networks be harnessed to enable a universal mechanism for topological isomerization? With this thought in mind, we hereafter illustrate such a TIN design and demonstrate its surprising benefit via construction of shape memory polymers (SMPs) with on demand shape-shifting versatility not offered by existing SMP.

## Results

**Network synthesis and mechanism of topological isomerization.** We use a readily accessible chemistry to ensure the general applicability of the TIN design. Specifically, our network was synthesized via photoinitiated radical polymerization between polyethylene glycol diacrylate (PEGDA, polyethylene glycol with $M_n$ of 3350) and N-hydroxyethylacrylamide (Fig. 1a). In this network, three features are noteworthy: polyethylene glycol (PEG) segments are crystallizable; PEGDA offers ester moieties; The N-hydroxyethylacrylamide comonomer provides pendent hydroxyls. In the presence of an organobase catalyst (1,5,7-triazabicyclo [4.4.0]dec-5-ene neutralized with two molar acetic acids, TBD), the ester bonds can be activated to undergo transesterification with the pendent hydroxyls. As a result, the network topology can isomerize according to the routes schematized in Fig. 1b, c.

The transesterification can occur between a hydroxyl group and an ester attached on a different main chain, namely inter-chain transesterification (Fig. 1b). Depending on the extent of the bond exchange, one or both ends of a PEG chain are released, resulting in isomeric states 2 and 3 (IS2 and IS3). Alternatively, the transesterification can happen between a hydroxyl group and an ester on the same main chain (Fig. 1c). The result of this intra-chain transesterification is that the one or both the PEG chain ends are also liberated, corresponding to IS4 and IS5, respectively. What is also intriguing is that, relative to the original isomeric state (IS1), the number of crosslinking points changes in different ways for the four isomeric states. The crosslinking points remain unchanged for IS2, but IS3 corresponds to doubling in crosslinking points. In contrast, the formation of intra-chain cycles for IS4 and IS5 leads to a reduction in the overall network crosslinking since the newly formed intra-chain cycles do not contribute to the crosslinking[31].

We note that Fig. 1b, c are simplified to demonstrate qualitatively the trend in increasing topological defects. In reality, a small amount of defects (intra-chain cycles, dangling chains, and free chains) are expected for the starting topology (IS1) as they are unavoidable for any real networks. These defects are negligible for simple demonstration only. In addition, the inter- and intra-chain transesterification can also occur in an inter-mixed fashion, resulting in more topologies beyond those shown in Fig. 1b, c. A more thorough discussion of this is provided in the supplementary information (see Supplementary Fig. 1 and related discussion). At this stage, direct and quantitative experimental characterization of the complex isomerization is difficult, although this is an interesting subject to study in the future, most likely via theoretical calculation. For the purpose of the current study, we focus hereafter on the statistical outcome in terms of the impact of topological isomerization on the number of crosslinking points and liberated PEG chains, or defects.

Overall, depending on the extent of dynamic bond exchange, the network isomerizes to a statistically mixed state (Fig. 1d) compromised of crosslink PEG chains (i), dangling PEG chains (ii), and free PEG chains (iii), inter-chain crosslink (iv), and intra-chain cycles (v). Of importance here is that the release of the PEG chain ends (ii and iii) introduces more segmental mobility that should favor the polymer crystallization. This is a mechanism that can be explored for programming thermomechanical properties of the network polymer. Additionally, the change in crosslinking offers an unusual opportunity in programming the rubbery modulus of the network material.

**Programming thermomechanical properties.** We anticipate that the number of hydroxyl groups would strongly affect the network topological isomerization. Accordingly, three network samples with different hydroxyl contents were synthesized by varying the amount of N-hydroxyethylacrylamide comonomer.

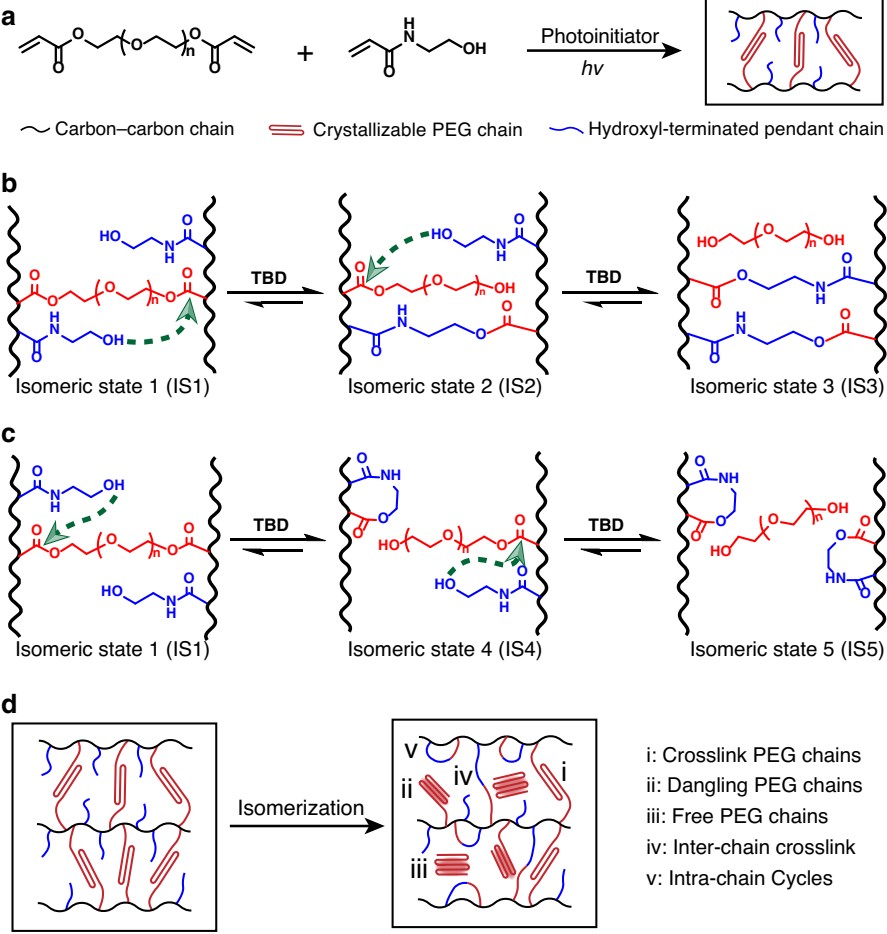

**Fig. 1 Design of the dynamic covalent network and the mechanism of topological isomerization. a** Polymer network synthesis. **b** Network isomerization via inter-chain transesterification pathways. **c** Network isomerization via intra-chain transesterification pathways. **d** Statistical outcome of the topological isomerization.

These samples were denoted as PEG-X, with X representing the hydroxyl/ester molar ratio. With the presence of TBD as the transesterification catalyst (2 wt%), all the samples were thermally annealed (120 °C, 60 min) for isomerization. The DSC results in Fig. 2a show that PEG-0, without the hydroxyl groups in the network, barely underwent any change in either melting temperature ($T_m$) or crystallinity ($X_c$). In contrast, both the $T_m$ and $X_c$ increase noticeably for PEG-1 and PEG-2, with the relative extent of increase being higher for PEG-2. We note that the starting PEG-2 has a slightly lower $T_m$ than PEG-1. This is because the hydroxyl groups affect the overall crystallization of the network. The as-synthesized PEG-0, PEG-1, and PEG-2 have similar gel fractions between 95.8% and 94.1%. After the thermal isomerization, the gel fractions for PEG-1 and PEG-2 reduce to 68.6% and 53.0%, respectively, whereas the gel fraction for PEG-0 remains unchanged (Supplementary Fig. 2). Furthermore, $^1$H NMR analysis verified the existence of free PEG chains, which are extracted from PEG-2 after thermal isomerization (Supplementary Fig. 3). The above results are consistent with the isomerization mechanisms outlined in Fig. 1b, c. When the hydroxyl/ester ratio is increased further to 3, the resulting PEG-3 is characterized and compared to PEG-2. The results (Supplementary Fig. 4) shows that a higher hydroxyl/ester ratio of 3 led to a lower melting temperature, with no obvious change in crystallinity. However, the melting temperature differences before and after isomerization remain almost identical for PEG-2 and

PEG-3. We hereafter focus on PEG-2 for further investigation in thermomechanical properties before and after the isomerization. We next resort to a photobase generator that can provide spatio-temporal release of a transesterification catalyst. Specifically, we employ a photobase generator that upon UV irradiation can release a strong organic base 1,5,7-triazabicyclo[4.4.0]dec-5-ene[29]. The catalyst amount can be controlled by the light irradiation time. Consequently, the thermally triggered bond exchange and the isomerization kinetics can be manipulated. Specifically, the thermal isomerization is conducted at 80 °C for 30 min hereafter. Accordingly, longer irradiation leads to network samples with progressively higher $T_m$ (Fig. 2b). Figure 2c shows that $X_c$ (calculated from DSC curves in Fig. 2b) increase with light irradiation, resulting in a similar increase for the storage modulus at 25 °C. When light irradiation is fixed at 300 s, Supplementary Fig. 5 suggests the crystallinity increases progressively with thermal annealing and reaches a plateau value at an annealing time of 30 min.

As for the gel fraction of an isomerized sample, it decreases with the irradiation time (Fig. 2d), consistent with Fig. 1b, c which suggest that the more extensive bond exchange results in more free PEG chains in the network. In determining gel fractions, a solvent washing step was required to remove the free chains in the network. Comparisons of the rubbery moduli before and after washing (Supplementary Fig. 6) can therefore shed more light on the isomerization process. From this figure, one can

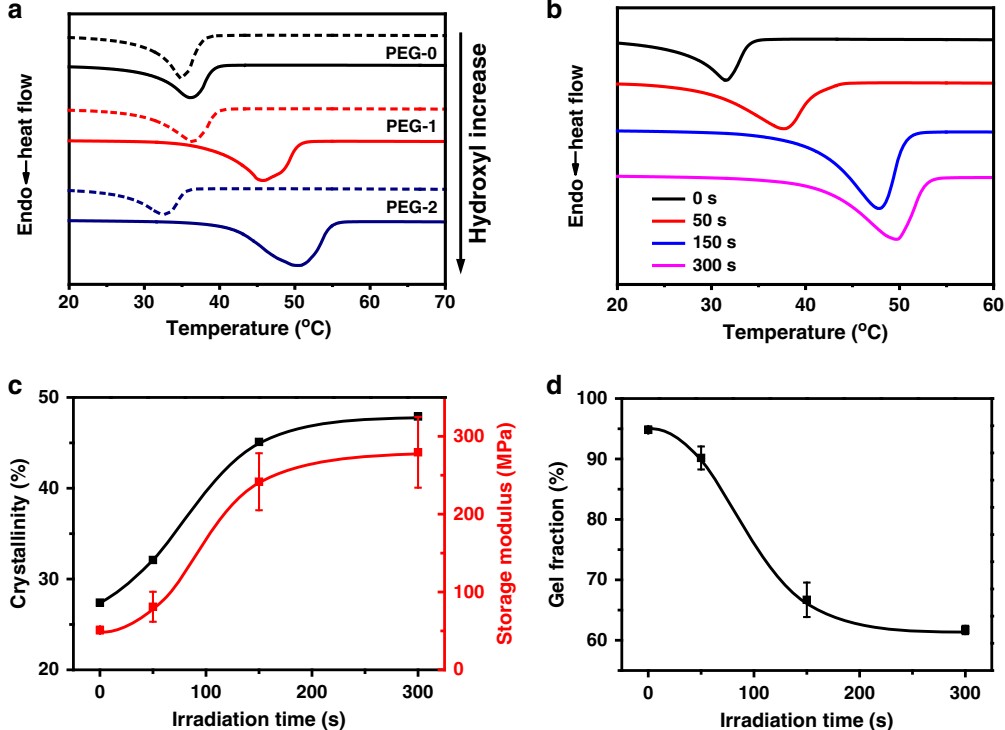

**Fig. 2 Programming thermomechanical properties via network topological isomerization. a** DSC curves of the network samples before (dotted lines) and after (solid lines) isomerization. **b** DSC curves for the isomerized PEG-2 samples obtained with different UV irradiation times. **c** Room temperature storage modulus (25 °C) and PEG crystallinities of the isomerized samples. **d** Gel fraction of the isomerized network samples. Error bars represent standard deviations calculated from three specimens.

see that the rubbery modulus of the isomerized material before washing decreases with irradiation time and reaches plateau values at irradiation time of 150 s. This is consistent with the formation of various topological defects (dangling chain, loops, and free chains) that do not contribute positively to the network. The rubbery modulus of the isomerized material after washing follows a similar trend but with absolute modulus values consistently higher than the corresponding samples before washing. This is a clear evidence that free chains produced during the isomerization indeed reduce the rubbery moduli.

**Spatio-selective topological isomerization.** The use of the photobase generator sets up the stage for spatio-selective control of the network isomerization and corresponding macroscopic properties. Figure 3a shows that the use of different photomasks allows control of the crystallization as reflected in the optical property change at different temperatures. At 80 °C, all the crystals melt and the films are transparent. At 0 °C, both the isomerized ($R_i$) and non-isomerized ($R_n$) regions crystallize, but the $R_i$ appears more opaque due to the higher crystallinity. When the temperature is raised to 37 °C, the contrast between the two regions become more evident due to the selective melting in the $R_n$ region. In addition to the optical properties, the modulus contrast between the isomerized and non-isomerized regions can readily switch. At 37 °C, the $R_n$ region is in its crystalline state and the $R_i$ region in its rubbery state. Consequently, the modulus of the former is 25 times that of the latter (Fig. 3b). At 80 °C, both regions are at their rubbery states and their modulus contrast is reversed. The $R_i$ region becomes 7 times softer than the $R_n$ region because of its lower crosslinking density and that the free PEG chains do not contribute to the modulus. This reversal in modulus contrast is visually demonstrated in Fig. 3c. At 37 °C, stretching is

mostly localized in the non-isomerized region, which is reversed at 80 °C. This unusual reversal in modulus may provide future opportunities in mechanically relevant multifunctional devices.

**Reversible and triple-shape memory performances.** The ability to spatio-selectively program network crystallization is particularly relevant for functional shape memory properties. Due to its crystalline nature, PEG-2 (before isomerization) shows reversible shape memory performance[32] when the temperature is switched between −20 and 80 °C, with 20% reversible strain under a constant stress of 0.8 MPa (Fig. 4a). The reversible strain is found to decrease linearly with the external stress (Fig. 4b), down to a negligible level when the stress is too small. To achieve stress-free reversible shape memory[33–36], current approaches rely on the introduction of an internal stress. This typically necessitates the use of two crystalline phases in which the high melting phase provides the internal stress whereas the low melting phase serves as the actuation phase[33–35]. In contrast to those macroscopically homogeneous microscopically phase separated systems, the ability to pattern macroscopic material heterogeneity via topological isomerization provides a unique mechanism to introduce internal stresses required for external stress-free reversible shape memory. The operating principle is illustrated in Fig. 4c. The spatially patterned sample is linearly stretched at 80 °C, above the melting temperatures of both $R_n$ and $R_i$ regions. Cooling down to 0 °C under the stretching force fixes the stretched shape. Upon removal of the force, the sample is heated to 37 °C. This selectively triggers the shape recovery in the $R_n$ region. This recovery is constrained by the $R_i$ region. The net effect is that the sample recovers to the 3D curved shape, corresponding to the stress balance between the two regions. In essence, internal stresses are created in both regions. When temperature is switched between

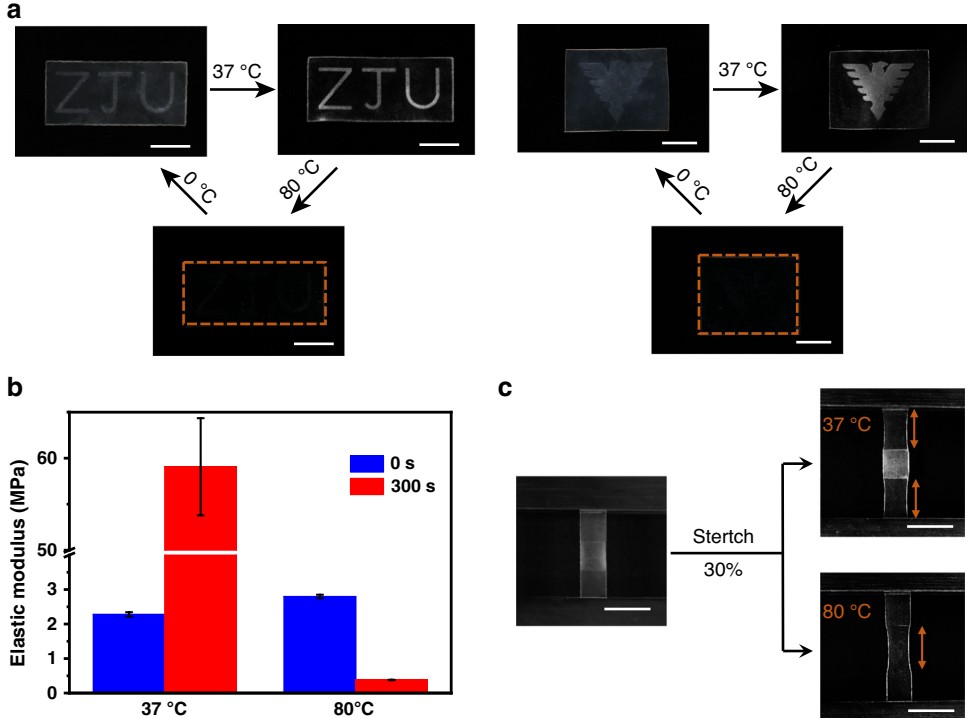

**Fig. 3 Spatio-selective topological isomerization. a** Temperature induced change in optical contrast due to the patterned isomerization/crystallization. The letters and logo correspond to the isomerized region with light irradiation of 300 s. The rest of the sample corresponds to non-isomerized region with no light irradiation. **b** Comparison of the elastic moduli for the non-isomerized and the isomerized network. **c** Reversal of modulus contrast at different temperatures. All scale bars: 10 mm and error bars represent standard deviations calculated from three specimens.

0 °C and 37 °C, the material shows reversible shape memory behavior with the low melting temperature $R_n$ serving as the actuation region. For the model pattern in Fig. 4c, the bending curvature can be easily tuned by changing the area ratio between $R_i$ and $R_n$ (Supplementary Fig. 7). The light patterns dictate the spatial stress within the film. Thus, employing diverse light patterns provide ample freedom to manipulate the reversible shape memory cycle (Fig. 4d). Importantly, the different geometric shapes involved in Fig. 4d are determined by the light patterns whereas the external deformation force is identical (i.e. simple linear stretching of 30%). This stands in sharp contrast to currently known reversible SMP for which the shapes involved are dictated by the external programming force. For those materials, to create the 3D shape shifting in Fig. 4d would require corresponding programming force much more complex than the linear stretching used for our material systems.

The opportunity to design the shape-shifting behavior on demand goes beyond the reversible shape memory. Figure 4e illustrates another possibility in manipulating triple-shape memory behavior. A polymer film with an isomerization pattern is subjected to linear stretching (30%) at 80 °C. Cooling down to 0 °C fixes this first temporary shape. Upon heating to 37 °C under stress-free condition, the film recovers to a second temporary 3D shape. Further heating to 80 °C recovers the original shape, the 2D film. Again, depending on the pattern, the second temporary 3D shape can be quite different (Fig. 4f). This triple-shape behavior has a key distinction with currently known triple-shape examples for which both temporary shapes are typically determined by the programming force(s)[37,38]. For the current system, while the first temporary shape is solely dependent on the programming force, the second temporary shape is dictated by both the pattern and programming force.

This characteristic brings up two unique benefits: A single programming force creates two temporary shapes that are non-linearly dependent; A simple stretching force allows access to complex 3D temporary shapes. Harnessing these attractive features for device applications may present interesting opportunities in the future.

## Discussion

In summary, we demonstrate a generally applicable molecular design of a topology isomerizable network (TIN) and show its unique versatility toward designable SMP on demand. Relying on the transesterification as the dynamic exchange reaction, this TIN can isomerize to various topological isomeric states with changes in topological defects. This in turn provides an opportunity to tailor the thermomechanical properties (crystallization and modulus). Coupled with spatio-temporal light regulation of the isomerization, the starting network polymer can evolve into many custom designable network composites. Based on this, we illustrate unusual opportunities in designing reversible SMP and multi-SMP with shape-shifting versatility beyond those reported elsewhere. At first glance, our approach may appear similar to what is achievable for two-stage polymerization in which network can be altered after the synthesis[4,39–41]. In reality, two-stage polymerization typically leads to higher crosslinking densities, without significant other changes. By comparison, many other network topological features can be tuned via topological isomerization. Consequently, the adjustable range of macroscopic properties is broader, as exemplified in Fig. 3b, c. On a broad basis, the concept of TIN greatly extends the design space for dynamic covalent networks with implications for many other types of adaptive polymers beyond SMP.

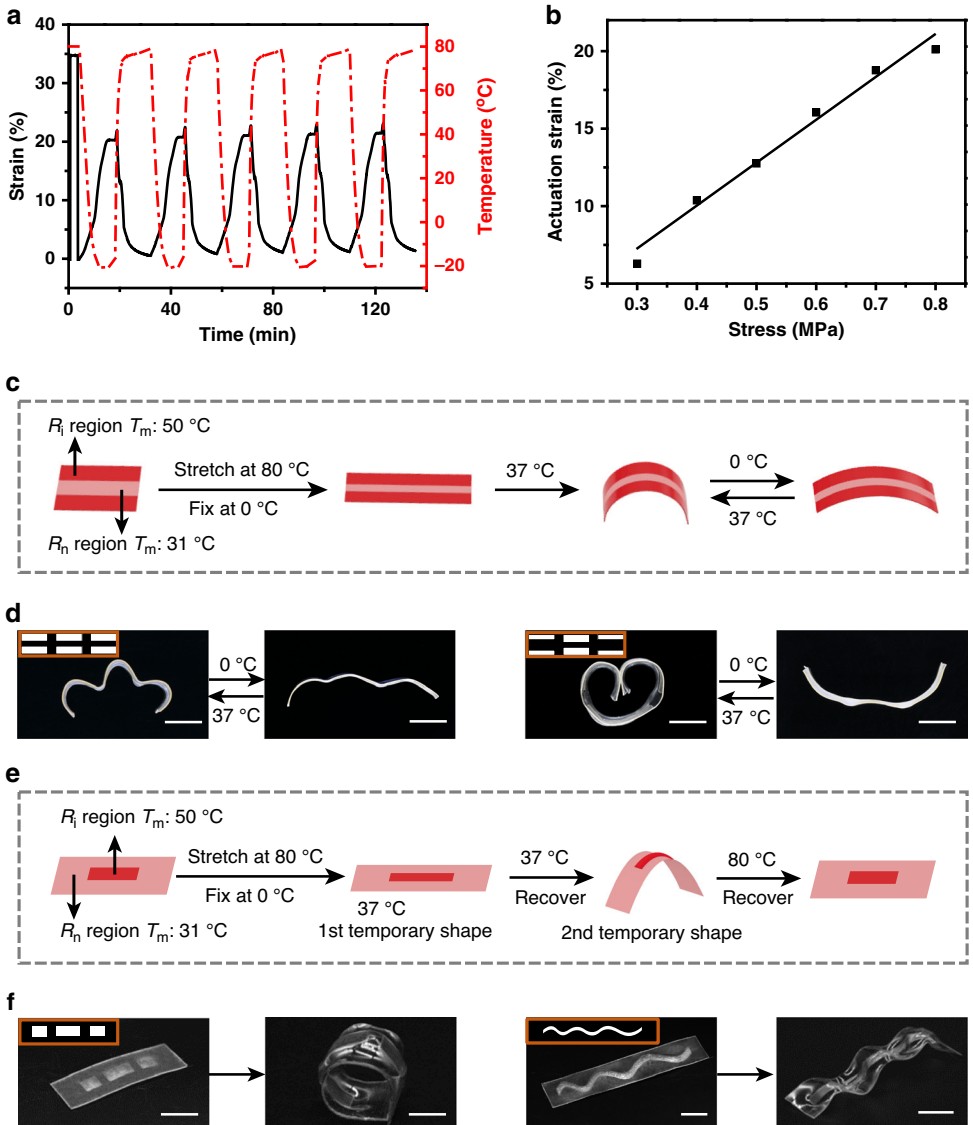

**Fig. 4 Reversible and triple-shape memory performances. a** Cyclic reversible actuation of a non-isomerized material under a constant stress (0.8 MPa).
**b** The relationship between reversible strain and the external stress. **c** Schematic illustration of reversible shape memory process. **d** Reversible shape
transformations obtained by patterned isomerization with the corresponding light patterns shown as the insets (light irradiation time: 300 s). **e** Schematic
illustration of the triple-shape memory process. **f** The original shape and 2$^{nd}$ temporary shape in the triple-shape memory cycles, with the corresponding
light patterns shown as the insets (light irradiation time: 300 s). (All scale bars: 10 mm).

## Methods

**Materials**. Polyethylene glycol ($M_n$ = 3350) was purchased from Sigma-Aldrich.
Irgacure 2959, N-hydroxyethylacrylamide, acryloyl chloride, trimethylamine, 1,5,7-
triazabicyclo[4.4.0]dec-5-ene, and ketoprofen were acquired from TCI. Acetic acid,
toluene, N,N'-dimethylformamide (DMF) were obtained from Guoyao chemical.
All chemicals were used as received. The photobase generator was synthesized
according to the literature[29].

**Synthesis of polyethylene glycol diacrylate (PEGDA)**. PEG (20.0 g, 5.97 mmol)
and trimethylamine (2.78 g, 27.50 mmol) were dissolved in toluene (200 mL).
Acryloyl chloride (2.49 g, 27.50 mmol) was added dropwise at 0 °C. The mixture
was kept at 80 °C for 20 h. The byproduct was removed by filtration and PEGDA
was obtained by precipitating the clear solution in hexane. The final product was
vacuum dried for 24 h at room temperature and its structure verified by $^1$H NMR
analysis (Supplementary Fig. 8).

**Synthesis of the polymer networks**. For the synthesis of PEG-2, PEGDA (0.5 g,
0.15 mmol) was dissolved in DMF (0.5 g) at 60 °C. N-hydroxyethylacrylamide
(0.07 g, 0.6 mmol) and photoinitiator (Irgacure 2959, 1.5 wt%) were added into the
PEGDA solution and stirred for several minutes. The homogenous mixture was

poured into a glass mold separated by a silicone rubber spacer (0.3 mm thickness)
and irradiated under UV light for 180 s (light source: IntelliRay 600 Flood UV,
intensity: 60 mW/cm$^2$). The obtained film was vacuum dried at 70 °C for 24 h, then
stored in a desiccator prior to testing. PEG-0 and PEG-1 were synthesized similarly
except that the amount of N-hydroxyethylacrylamide was varied according to the
hydroxyl/ester molar ratio.

**Introduction of the photobase generator (PBG)**. PBG was introduced into a
polymer network by soaking the material in 2 wt% PBG toluene solution for
20 min. With this procedure, 6 wt% of PBG (calculated from mass change) was
incorporated into the network polymer. For light irradiation experiments, a
polymer sample was irradiated equally on both sides and the irradiation time was
taken as the sum of exposure times on both sides.

**Gel fraction tests**. A specimen was weighed before soaking in CHCl$_3$ for 24 h
(replacing the solvent every 12 h). The swollen specimen was then vacuum dried
for 24 h at room temperature and weighed. The gel fraction was calculated based
on the ratio between the final and initial weights.

**Characterization**. Differential scanning calorimetry (DSC) was conducted using aTA Q200 instrument. All samples were first equilibrated at 80 °C and cooled to −50 °C at a rate of 5 °C/min. The DSC curves were obtained in a second heating run at a rate of 5 °C/min. Dynamic thermomechanical analysis (DMA) was conducted using aTA Q800 instrument. The reversible shape memory curves were obtained in a "controlled force" mode. The storage modulus was obtained in a "multi-frequency strain" mode with an amplitude of 20 μm at 1 Hz. Elastic moduli were determined by tensile tests conducted with a Zwick/Roell Z005 machine under a tensile speed of 10 mm/min. A minimum of three specimens in a standard dumbbell shape were tested.

## Data availability

The data that support the findings of this study are available from the corresponding author on reasonable request.

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

## Acknowledgements

This work was supported by the following programs: National Natural Science Foundation of China (No. 21625402, 51822307 and 51673169). The authors also thank Mrs. Li Xu for her assistance in performing DSC analyses at State Key Laboratory of Chemical Engineering (Zhejiang University).

## Author contributions

T.X. conceived the concept. W.M. designed and conducted the experiments. W.M. and T.X. wrote the paper. W.M., W.Z., B.J., N.Z., and Q.Z. analyzed experimental results. W.M., W.Z., B.J., C.N., N.Z. Q.Z., and T.X. contributed to the discussion.

## Competing interests

The authors declare no competing interests.
