## [Peer Review File · Nature Communications]

REVIEWER COMMENTS

Reviewer #1 (Remarks to the Author):

This study applies the concept of topological isomerization to programming the shape memory behavior of polymer networks by introducing internal stresses to enable reversible shape memory without applying external stress. Distinct features of this approach include light triggered regulation of polymer topology within a solid material without using external reagents, which significantly expands the range of practical applications. Within this general concept, the authors address a specific problem of harnessing topological defects to control shape, which has very exciting implications for establishing a universal mechanism of topological isomerization. The paper is recommended for publication after minor revisions outlined below:

1) I am skeptical about using the word adaptive in the first and fourth sentences of the introduction, unless the authors explain its meaning in the context of shape memory. At low resolution, refs 1-8 describe different kinds of stimuli-responsive behaviors, including shape memory. The authors may want to continue sentence 4 as:

Indeed, dynamic covalent polymer network exhibits unusual adaptive properties not present in typical covalent networks such as...TBC [ref].

2) Also, the term network topology should be clearly defined as it is the main focus of this paper. The authors may want to continue the following sentence as:

A counterintuitive opportunity arises when a dynamic covalent network is designed such that bond rearrangement leads to different topologies such as TBC...

3) The authors mentioned that topological defects such as loops affect macroscopic properties. What properties are we talking about? How significant is their effect? At small fraction of such defects the effect is actually negligible. What fractions of topological defects is produced in these systems?

4) In page 7 and Figure 2d, it should be gel fraction (not content).

5) PEG melting temperature depends on molecular weight. What is MW range of the PEG sections (both tethered and deliberate)?

6) PEG is a hydrophilic polymer. Therefore, all properties including thermal, mechanical, and shape memory depend on water sorption. Please comment.

Reviewer #2 (Remarks to the Author):

On Demand Shape Memory Polymer via Light Regulated Topological Defects in a Dynamic Covalent Network

The authors explored covalent adaptable networks (CANs) crosslinked through UV curing of PEG-diacrylate and N-hydroxyethylacrylamide to design topology isomerizable networks (TINs) with varying amount of free hydroxyls. The cured networks were then immersed in a toluene solution

containing photolabile protected from 1,5,7-triazabicyclo[4.4.0]dec-5-ene (TBD, 2 wt%). The dynamic nature of the networks proceeded through transesterification catalyzed by TBD, after photoliberation, with free hydroxyl groups along the backbone. The authors show that, after irradiation, the hydroxyl groups can liberate PEG crosslinks causing an increase in the crystallinity. This spatiotemporal control was used for the generation of shape memory materials. The authors show an increase in crystallinity of the networks by differential scanning calorimetry (DSC) and dynamic mechanical analysis (DMA) after irradiation of UV light, attributed to the liberation of PEG crosslinks. This liberation of PEG is also observed by a decrease in the gel content over time by gel sol studies and soluble fraction containing PEG. The authors also show that without the presents of irradiation or hydroxyl groups the isomerization (loss of PEG) does not happen. The concept and results (effecting network crystallinity through light-induced transesterification by TBD) are very similar to their recent publication titled "Light-triggered topological programmability in a dynamic covalent polymer network" where "TIN" was introduced [Sci. Adv. 2020; 6: DOI eaaz2362]. Although this application of defects in covalent adaptable networks for shape memory materials is an interesting topic, I do not believe this study to be impactful enough for Nat. Commun. For these reasons and the comments below, I believe this manuscript should be published in a more specific journal and requires additional experimental work.

Questions/Concerns:

(1) Temperature sweeps of networks at various irradiation times/washed samples by DMA should be performed to show a change in the rubbery plateau. The rubbery plateau gives information about the molecular weight between crosslinks and will give more insight to change in network topology.

(2) Kinetic studies for the consumption on the two monomers, where PEG-acylate substitutes PEG-diacylate, should be performed to show the composition of the polymer backbones (statistical, gradient, vs homopolymers).

(3) FTIR at all stages of the network would be nice to show the preservation of the network throughout the sample.

(4) Some of the writing could be reworded for clarity and improved flow:

e.g., (not comprehensive)

(1) The word "unusual" is used quite often and it could be replaced as seems misleading to say this effect is rare (covalent adaptable networks have become quite common in the literature over past two decades).

(2) Page 2, line 6: "Dynamic covalent bonds, with their unique combination of adaptability and high bond strength, have the potential to shift the paradigm." This sentence seems unfinished.

(3) Page 2, line 13: "Triggering the dynamic characteristics of the covalent bonds, however, renders the network polymers malleable through covalent bond exchange within the network. With rheological behaviors similar to inorganic glass, this class of network polymers (vitriimer) can be mold reprocessed." These sentences are disjointed and unrelated. Worse this leads the reader to believe that all covalent adaptable networks are vitrimers. This is not the case and should be reworded.

(4) Page 2, line 19: I don't believe this sentence is necessarily true, "Although it [solid-state plasticity] relies on the same mechanism of network bond rearrangement, the equilibrium for the dynamic bond exchange does not need to be pushed to a level required for vitrimers." Vitrimers are a specific class of covalent adaptable networks and solid-state plasticity can be achieved through several types of CANs.

(5) Line 4 in the conclusion states, "This in return..." and should be "This in turn..."

(5) Moles or mole ratios should be given for experimental procedure.

(6) Both NMR figures should be more legible in ESI and integrations for PEG-diacrylate would be nice.

Reviewer #3 (Remarks to the Author):

In this work, the authors demonstrate a molecular design of a topology isomerizable network. By leveraging the BERs between dangling chains and mechanically effective chains, the network thermomechanical properties (crystallinity and modulus) can be readily tailored. This unique mechanism was further extended to design shape-changing polymers. Overall, I enjoyed reading the paper and believe it provides valuable insights to the community to design innovative active polymers. I suggest accepting the paper with the following minor comments:

1. The relationships between heating time and isomerization degree should be characterized. Does the crystallinity keep increasing progressively or eventually saturate at some level?
2. How will it change when more hydroxyl group presents in the network., namely the hydroxyl/ester ratio is great than 2? The authors are suggested to elaborate or characterize.
3. When was the TBD added during the network synthesis? UV light was used to polymerize the network, did that release the BER catalyst, and thus affect the network properties?
4. There are existing studies (e.g., [10.1038/s41467-018-04292-8](https://doi.org/10.1038/s41467-018-04292-8), [10.1016/j.jmps.2019.02.013](https://doi.org/10.1016/j.jmps.2019.02.013)) using two-stage polymerization to tailor the network crosslinking degree and thermomechanical properties, which are similar to the authors' approach. The authors are suggested to discuss and cite the works in the manuscript.

Reviewer #1

This study applies the concept of topological isomerization to programming the shape memory behavior of polymer networks by introducing internal stresses to enable reversible shape memory without applying external stress. Distinct features of this approach include light triggered regulation of polymer topology within a solid material without using external reagents, which significantly expands the range of practical applications. Within this general concept, the authors address a specific problem of harnessing topological defects to control shape, which has very exciting implications for establishing a universal mechanism of topological isomerization. The paper is recommended for publication after minor revisions outlined below:

Response: We thank the reviewer for the positive comments and constructive suggestions that help us improve the paper.

1. I am skeptical about using the word adaptive in the first and fourth sentences of the introduction, unless the authors explain its meaning in the context of shape memory. At low resolution, refs 1-8 describe different kinds of stimuli-responsive behaviors, including shape memory. The authors may want to continue sentence 4 as:

Indeed, dynamic covalent polymer network exhibits unusual adaptive properties not present in typical covalent networks such as....TBC [ref].

Response: Agreed. In the first sentence, “Adaptive” was changed to “Stimuli-responsive”. The fourth sentence was changed to “*Indeed, dynamic covalent polymer network exhibits adaptive properties not present in typical covalent networks such as self-healing, reprocessing, and permanent shape reconfiguration.*”

2. Also, the term network topology should be clearly defined as it is the main focus of this paper. The authors may want to continue the following sentence as:

A counterintuitive opportunity arises when a dynamic covalent network is designed such that bond rearrangement leads to different topologies such as TBC...

Response: Agreed. The sentence has been changed to “*A counterintuitive opportunity arises when a dynamic covalent network is designed such that bond rearrangement leads to different topologies such as crosslinking densities, physical entanglements, and dangling chain length/distribution.*”

3. The authors mentioned that topological defects such as loops affect macroscopic properties. What properties are we talking about? How significant is their effect? At

small fraction of such defects the effect is actually negligible. What fractions of topological defects is produced in these systems?

Response: This particular sentence on page 4, line 3 was revised to “*These defects affect the macroscopic mechanical and thermo-mechanical properties*”. As the reviewer correctly recognized, the significance of the effect depends quantitatively on the level of defects and it can indeed be negligible at small fractions. Our objective was NOT to quantify the topological defects, this was the subject of several nice papers referenced (30, 31). The goal here was to manipulate the relative change of these defects and how the change impacts the macroscopic properties. One of the topological defects we did measure quantitatively is gel fractions (see Fig. 2d).

4. In page 7 and Figure 2d, it should be gel fraction (not content).

Response: “gel content” was changed “gel fraction” throughout the manuscript.

5. PEG melting temperature depends on molecular weight. What is MW range of the PEG sections (both tethered and deliberated)?

Response: The molecular weight of PEG is 3350, which is now highlighted in the main text. Topological isomerization does not affect the PEG main chain, thus its molecular weight remains unchanged.

6. PEG is a hydrophilic polymer. Therefore, all properties including thermal, mechanical, and shape memory depend of water sorption. Please comment.

Response: Indeed, PEG is a hydrophilic polymer and can gradually absorb water in the air. In our experimental process, all synthesized materials were vacuum-drying at 70 °C for 24 h to remove solvent and free water, then stored in a desiccator prior to testing. Thus, the effect of ambient moisture was minimized. This information is now added in the experimental section.

Reviewer #2

The authors explored covalent adaptable networks (CANs) crosslinked through UV curing of PEG-diacrylate and N-hydroxyethylacrylamide to design topology isomerizable networks (TINs) with varying amount of free hydroxyls. The cured networks were then immersed in a toluene solution containing photolabile protected from 1,5,7-triazabicyclo[4.4.0]dec-5-ene (TBD, 2 wt%). The dynamic nature of the networks proceeded through transesterification catalyzed by TBD, after photoliberation, with free hydroxyl groups along the backbone. The authors show that, after irradiation, the hydroxyl groups can liberate PEG crosslinks causing an increase in the crystallinity. This spatiotemporal control was used for the generation of shape memory materials. The authors show an increase in crystallinity of the networks by differential scanning calorimetry (DSC) and dynamic mechanical analysis (DMA) after irradiation of UV light, attributed to the liberation of PEG crosslinks. This liberation of PEG is also observed by a decrease in the gel content over time by gel sol studies and soluble fraction containing PEG. The authors also show that without the presents of irradiation or hydroxyl groups the isomerization (loss of PEG) does not happen. The concept and results (effecting network crystallinity through light-induced transesterification by TBD) are very similar to their recent publication titled “Light-triggered topological programmability in a dynamic covalent polymer network” where “TIN” was introduced [Sci. Adv. 2020; 6: DOI eaaz2362]. Although this application of defects in covalent adaptable networks for shape memory materials is an interesting topic, I do not believe this study to be impactful enough for Nat. Commun. For these reasons and the comments below, I believe this manuscript should be published in a more specific journal and requires additional experimental work.

Response: We thank the reviewer for his/her appreciation of controlling topological defects in a dynamic covalent network. We understand the concern of impact given our prior publication on TIN, although we believe the focus of the current work is sufficiently different as explained below. In our prior work, the network design is very niche based, implying that TIN is not necessarily a generally applicable concept. If that is the case, the appeal of TIN would be very narrow, perhaps limited to just a few papers. Our current work shows that TIN is a generally applicable concept by simply viewing the often forgotten (and undesirable) defects as tunable topological features. I believe this would surprise many who have been working on the subject. More importantly, this could open up many more opportunities for fundamental exploration of the molecular designs

and technological innovations due to the broad applicability. Let me emphasize again that this is NOT a simple extension of the prior work or simply an alternative molecular design, which would be incremental and therefore unsuitable for Nat. Commun. Instead, this is a story that says “hey, most polymer networks can be designed to have these attractive features”. Based on the above arguments, we believe our current work is going to be very impactful, in ways that are different from our prior work. We are glad that other reviewers seem to recognize this given that they are also aware of this prior publication.

Many branches of polymer science face similar situations. Taking self-healing polymers as an example, the original concept was proposed long time again (not sure when), yet the field still faces many challenges. This is why many high profile papers have been published on the same general topic. It is my personal belief that, if a field is sufficiently important, it takes many great works/papers to push it ahead. We don't feel the potential impact of the current work is reduced by our prior work. Instead, the two pieces of work would enhance each other given their very different focuses on the same subject.

Questions/Concerns:

1. Temperature sweeps of networks at various irradiation times/washed samples by DMA should be performed to show a change in the rubbery plateau. The rubbery plateau gives information about the molecular weight between crosslinks and will give more insight to change in network topology.

Response: This is a great suggestion that would indeed shed more light on the mechanism. We've conducted additional experiments accordingly and the results are presented in Supplementary Fig. 6. From this figure, we can see that the rubbery modulus of the isomerized material before washing decreases with irradiation time and reaches plateau values at irradiation time of 150 s. This is consistent with the formation of various topological defects (dangling chains, loops, and free chains) that do not contribute positively to the network. The rubbery modulus of the isomerized material after washing follows a similar trend but with absolute modulus values consistently higher than the corresponding samples before washing. This is a clear evidence that free chains produced during the isomerization indeed reduce the rubbery moduli. The above description and discussion are now presented on paragraph 2 of page 8.

2. Kinetic studies for the consumption on the two monomers, where PEG-acrylate substitutes PEG-diacrylate, should be performed to show the composition of the polymer backbones (statistical, gradient, vs homopolymers).

Response: We believe there is a misunderstanding here. No PEG-acrylate is used in this work. Instead, the two monomers used are PEG-diacrylate and N-hydroxyethylacrylamide (Fig. 1a). For network formation, gel point is typically reached at a high monomer conversion. At different polymerization times before reaching the gel point, the incorporation of the two monomers into polymer chains may differ due to the different monomer reactivities. Correspondingly, the system is a statistical mixture of polymer chains of different compositions. When gel point is reached, these polymer chains are linked randomly into a network. The above discussion suggests that the network composition is statistical. In principle, we can prove with kinetic studies that the two monomers will react differently, but it does not change the above discussion and the network has to be statistical. Practically, it is also difficult to conduct the kinetic studies because the polymerization is too fast (within seconds) and we do not have access to real-time FTIR, especially given the current pandemic.

3. FTIR at all stages of the network would be nice to show the preservation of the network throughout the sample.

Response: According to the isomerization mechanism, the chemical composition of the network does not change during the process, only the way the bonds are connected changes. Based on this mechanism, FTIR would not detect any changes. No revision is made here.

4. Some of the writing could be reworded for clarity and improved flow:

e.g., (not comprehensive)

(1) The word “unusual” is used quite often and it could be replaced as seems misleading to say this effect is rare (covalent adaptable networks have become quite common in the literature over past two decades).

Response: Page 1 line 2, “unusual” has been replaced as “many”. Page 2 line 9, “unusual” has been deleted.

(2) Page 2, line 6: “Dynamic covalent bonds, with their unique combination of adaptability and high bond strength, have the potential to shift the paradigm.” This

sentence seems unfinished.

Response: The sentence has been changed to “*Dynamic covalent bonds, with their unique combination of adaptability and high bond strength, can be a paradigm shift*”.

(3) Page 2, line 13: “Triggering the dynamic characteristics of the covalent bonds, however, renders the network polymers malleable through covalent bond exchange within the network. With rheological behaviors similar to inorganic glass, this class of network polymers (vitriimer) can be mold reprocessed.” These sentences are disjointed and unrelated. Worse this leads the reader to believe that all covalent adaptable networks are vitrimers. This is not the case and should be reworded.

Response: Completely agreed and sorry for the confusion. The word “malleable” was changed to “reprocessable”. This following statement is deleted altogether to avoid the confusion “With rheological behaviors similar to inorganic glass, this class of network polymers (vitriimer) can be mold reprocessed”.

(4) Page 2, line 19: I don’t believe this sentence is necessarily true, “Although it [solid-state plasticity] relies on the same mechanism of network bond rearrangement, the equilibrium for the dynamic bond exchange does not need to be pushed to a level required for vitrimers.” Vitrimers are a specific class of covalent adaptable networks and solid-state plasticity can be achieved through several types of CANs.

Response: Agreed. Since this sentence is not essential for this manuscript, we decided to delete it to avoid the confusion.

(5) Line 4 in the conclusion states, “This in return...” and should be “This in turn...”

Response: Revised accordingly.

(6) Moles or mole ratios should be given for experimental procedure.

Response: Revised accordingly.

(7) Both NMR figures should be more legible in ESI and integrations for PEG-diacrylate would be nice.

Response: More legible ¹H NMR figures and integrations for PEG diacrylate have been provided in the supplementary information.

Reviewer #3

In this work, the authors demonstrate a molecular design of a topology isomerizable network. By leveraging the BERs between dangling chains and mechanically effective chains, the network thermomechanical properties (crystallinity and modulus) can be readily tailored. This unique mechanism was further extended to design shape-changing polymers. Overall, I enjoyed reading the paper and believe it provides valuable insights to the community to design innovative active polymers. I suggest accepting the paper with the following minor comments:

Response: We thank the reviewer for the positive comments and useful suggestions below.

1. The relationships between heating time and isomerization degree should be characterized. Does the crystallinity keep increasing progressively or eventually saturate at some level?

Response: Additional experiments were conducted and the results are presented in Supplementary Fig. 5. As expected by the reviewer, the crystallinity increases progressively and reaches a plateau value at an annealing time of 30 minutes. The description is added in the main text of page 8, paragraph 1.

2. How will it change when more hydroxyl group presents in the network., namely the hydroxyl/ester ratio is great than 2? The authors are suggested to elaborate or characterize.

Response: We further investigated PEG-3, for which hydroxyl/ester ratio is 3. In comparison with PEG-2 (supplementary Fig. 4), a higher hydroxyl/ester ratio of 3 led to a lower melting temperature, with no obvious change in crystallinity. In addition, the melting temperature difference before and after isomerization remains almost identical for PEG-2 and PEG-3. The related discussion is now provided in the main text of page 7.

3. When was the TBD added during the network synthesis? UV light was used to polymerize the network, did that release the BER catalyst, and thus affect the network properties?

Response: TBD was added into the monomers before the UV polymerization. It does not affect the polymerization. For the photobase generator (PBG) that generates TBD upon light irradiation, it was added after the network synthesis by

solution swelling and drying. UV light that triggers the polymerization does not affect the BER release, nor does it impact the network properties. This description is presented in the experimental section of page 13.

4. There are existing studies (e.g., 10.1038/s41467-018-04292-8, 10.1016/j.jmps.2019.02.013) using two-stage polymerization to tailor the network crosslinking degree and thermomechanical properties, which are similar to the authors' approach. The authors are suggested to discuss and cite the works in the manuscript.

Response: These two references, along with another paper from Bowman's group, are added as references 39, 40, 41. The comparison with two-stage polymerization is made in the Discussion section as follows: “*At first glance, our approach may appear similar to what is achievable for two-stage polymerization in which network can be altered after the synthesis. In reality, two-stage polymerization typically leads to higher crosslinking densities, without significant other changes. By comparison, many other network topological features can be tuned via topological isomerization. Consequently, the adjustable range of macroscopic properties is broader, as exemplified in Figs. 3b and 3c.*”

REVIEWERS' COMMENTS:

Reviewer #1 (Remarks to the Author):

The authors have completely and thoroughly replied to my comments. The paper is recommended for publication.

Reviewer #3 (Remarks to the Author):

the manuscript is suggested to be accepted